**Perspective**

# A framework for neurophysiological experiments on flow states
Oliver Durcan [1][✉], Peter Holland [1,2] & Joydeep Bhattacharya[1,2]

Csikszentmihalyi's concept of the "flow state" was initially discovered in experts deeply engaged in self-rewarding activities. However, recent neurophysiology research often measures flow in constrained and unfamiliar activities. In this perspective article, we address the challenging yet necessary considerations for studying flow state's neurophysiology. We aggregate an activity-autonomy framework with several testable hypotheses to induce flow, expanding the traditional "challenge skill balance" paradigm. Further, we review and synthesise the best methodological practices from neurophysiological flow studies into a practical 24-item checklist. This checklist offers detailed guidelines for ensuring consistent reporting, personalising and testing isolated challenge types, factoring in participant skills, motivation, and individual differences, and processing self-report data. We argue for a cohesive approach in neurophysiological studies to capture a consistent representation of flow states.

The "flow state" was initially conceptualised in 1975 by Csikszentmihalyi[1] in his attempt to understand the essence of the "optimal experience" or what people commonly describe as being "in the zone". Through interviews with hundreds of experts in a broad range of activities, Csikszentmihalyi[1] noticed a consistent pattern in their subjective accounts of flow. For example, expert rock climbers described the experience as if "you're moving in harmony with something else, you're a part of it" (p. 81); dancers: "once I get into it, then I just float along, having fun, just feeling myself move around" (p. 104), and writers: "it is really the fingers that are doing it and not the brain. Sometimes the writing takes charge" (p. 118). After 50 years of conceptual refinement, flow is defined as "an intrinsically rewarding state of absorption in a task in which control feels effortless" ([2] p. 819). There are a growing number of experimental studies that investigate the neurophysiological correlates of the flow state[3–5]. However, of the experiments conducted so far, many activity contexts and methodological approaches to inducing flow states remain unexplored.

Approaches to experimentally induce flow states have seen very little variation since the concept of flow was introduced. Flow states are commonly induced by balancing participants' skills with challenges—tasks that are neither too easy nor too hard—and ensuring these tasks have clear goals and immediate, unambiguous feedback[6,7]. The measured outcomes are then compared to control conditions, which create an imbalance of challenges and skills (either too easy or too hard). Whilst the challenge skill balance is widely used, it is not the only antecedent of flow. Other important factors include autonomy (the freedom to choose and engage in the activity[8–12]), self-efficacy (the perceived ability to manage the activity[13–17]), skills[18–21], and

perception of the activity as important[22,23], interesting[7,24,25] and intrinsically motivating (self-rewarding)[8,11,25–27]. Flow is not just dependent on a balance between challenges and skills, despite many research designs relying on this single antecedent[28–30]. Individuals must also have some subjectively inclined, motivated relationship with the activity to experience flow. However, systematically fulfilling these conditions in experiments testing the neurophysiological correlates of flow states is rare[4].

The reason motivation-related antecedents are not met in these experiments may lie in the restrictions of neurophysiological measures. A main limitation of neurophysiological methods is the use of sterile laboratory environments[31–33], which has been unavoidable due to the lack of high-quality but portable neural and physiological data collection equipment[6]. Moreover, neural and physiological signals are prone to movement artefacts, restricting participants' movement and creating an unnatural environment. Considering the multi-dimensional nature of flow, research designs have also been restricted by statistical methods that can analyse only a small range of experimental conditions (e.g., three levels of challenge). It is only recently that advances in movement-permitting imaging equipment[34] and statistical methods[35] have become available, which is promising for future flow research.

To overcome the limitations associated with standard neural and physiological data collection techniques and statistical methods, this flow research has often relied on rudimentary experimental activities and procedures. The most common experimental activities are arcade games like Tetris, Pac-Mac, and Asteroid Impact (e.g.,[28,36,37]) and math activities (e.g.,[29]). In other studies, first-person shooter (FPS) games like Half-Life 2,

[1]Department of Psychology, Goldsmiths University of London, London, UK. [2]These authors jointly supervised this work: Peter Holland, Joydeep Bhattacharya.
[✉]e-mail: odurc001@gold.ac.uk

Tactical Ops: Assault on Terror, and Call of Duty: Modern Warfare 2 have been adopted to study flow (e.g.,[38–40]). The duration of game engagement in these studies is often short, lasting an average of around 12 min[41]. These trends generally represent a limited variety of activities that are not autonomy-supportive, interesting, important, or intrinsically motivating for participants.

Despite these limitations, valuable and wide-ranging insights have been gained through extensive experimentation using the challenge skill balance paradigm. Several review articles have explored the involvement of various systems such as reward and dopamine, attention and executive control, default mode network[3–5,41–43], cardiovascular, electrodermal, muscular, cortisol[3,5,41], cerebellar[5,41,43], respiratory, motor, optical[5,41], multiple demand[3,5], salience, and locus coeruleus-norepinephrine[42,43] systems (also see[44–48]). However, these reviews consistently highlight a lack of consensus in the reported findings of neurophysiological studies on flow states. Manipulating the challenge skill balance antecedent has resulted in both supporting and contradicting theoretical perspectives, including Weber et al.'s[49] proposal of co-activation of attention and reward networks and Dietrich's[50] proposal of downregulation of the executive attention network during flow[4]. Physiological findings are also inconsistent, with varying activation patterns of the parasympathetic and sympathetic nervous systems across the challenge spectrum[41]. Furthermore, cortisol levels and facial muscular activations have linear positive, linear negative, inverted-U shaped, and non-existent associations with challenges between studies[5,41].

Methods to reliably induce flow states in neurophysiological experiments are still being explored. Whilst these activity and procedural trends may capture certain aspects of flow states, they do not represent their full range of characteristics. A more comprehensive observation came from Csikszentmihalyi and colleagues' original flow research, where participants were usually experts who had invested a large portion of their lives in their respective, self-determined, and routinely practised activities[1,51–54]. When Csikszentmihalyi asked participants about their reasons for engaging in these activities, their responses often centred around enjoyment, skill utilisation, and personal development, reflecting intrinsic motivation. While subsequent flow research has emphasised challenge skill balance as a primary flow antecedent, the role of skill level and motivation has received comparatively less attention. Empirical testing has demonstrated a positive association between these variables and flow outcomes, indicating their instrumental role in facilitating the flow state rather than being merely coincidental (intrinsic motivation[2,8,25,26]; self-efficacy[14–16]; skill level[18,19,55]; self-initiated engagement[10–12]). Csikszentmihalyi did not explicitly label motivational factors as flow antecedents, but their presence in his early studies and absence in many recent neurophysiological studies is discernible.

We argue that the sufficient requirements for flow to emerge might not be met in contemporary neurophysiological flow experiments. This discrepancy might account for the inconsistent experimental outcomes, raising the questions: Are flow states being induced properly in these experiments? Or are different states being measured, erroneously assumed to represent flow? To encourage a wider variety of experimental activities in neurophysiological flow studies, we present a testable activity-autonomy framework. To address methodological approaches for testing skilled, motivated participants, we present a methodological checklist, aimed at promoting the adoption of best practices in future studies on flow states. We suggest that properly assessing this methodology would be crucial to ensure that the induced states align with the concept of flow.

## Challenge types and activity autonomy

The ability to make domain-general claims about the neurophysiological correlates of flow states is currently hindered by several factors. The range of experimental activities used to induce and measure flow is limited in the studies conducted so far[4]. Without measuring and comparing flow across diverse contexts, it is uncertain if they all produce identical neurophysiological correlates of flow states. Additionally, experiments often compare challenge levels instead of challenge types to make inferences about flow. While distinguishing high or low challenge levels is straightforward,

differentiating challenge types in various activities is less so. For instance, the challenges posed by music improvisation are distinct from those in mental arithmetic, yet it is assumed they yield the same flow state if balanced with skills. A systematic method to identify both similar and distinct characteristics of activities is crucial to compare their effects on flow states and their neurophysiological correlates. Therefore, we propose a framework that can be used by researchers to distinguish experimental activities based on their autonomy characteristics (Fig. 1).

Autonomy, also known as self-determination[56–58], refers to individuals' choice to self-initiate engagement in an activity. For example, a child may play football because they want to (self-determined or autonomous) or because it is part of their school curriculum (determined by external reasons or controlled)[51,56]. Autonomy has been consistently linked to flow states, with studies indicating a positive association[11,12,59]. Csikszentmihalyi found that autonomy-supportive activities, such as composing music, were associated with higher intrinsic rewards than rule-bound activities like chess (see Table 3 in[1], p. 31). Consistent with this, a cross-sectional survey of 1709 adults found more frequent flow states in creative activities compared to sports activities[60]. In a systematic review of flow in adventure recreation activities, qualitative studies showed that feeling pressure to hurry inhibits flow[61]. Furthermore, a meta-analysis of flow state research showed that challenge skill balance correlated with flow more than twice as highly in leisure activities compared to work or education contexts[30]. Since neurophysiological flow studies typically adopted autonomy-controlling activities (e.g.,[28,29,62]), they may not have set up the most conducive conditions for flow states.

Figure 1 presents four spectrums that can be used to distinguish isolated characteristics of activities. Interaction speeds can range from relaxed (permitting temporal freedom) to pressured (demanding instant responses). The feedback used to steer activity progress can range from permitting subjective interpretation (like intuition, inspiration, or emotional response) to being strictly objective (coming from the stimuli only). The range of strategies used to progress towards activity goals can range from open (decided by the individual) to fixed (restricted by the activity). Finally, the activity goals can be open (changeable and open to interpretation) or fixed (prescribed and inflexible). Activities may represent a narrow (e.g., Tetris, red area in Fig. 1) or wide (e.g., abstract painting, blue area in Fig. 1) autonomy range since some activities have more customisable and flexible structures than others. Additionally, the way individuals engage in the same activities can vary significantly across different contexts. For example, creating an abstract painting may be autonomy-controlling in one context and autonomy-supportive in another. Notably, many successful painters hire assistants to create their artworks according to specific rules[63], resulting in a highly regulated painting process. These variations highlight the importance of recognising distinct autonomy characteristics in unique activity contexts, which likely imply distinct neurophysiological signals. Merely reporting the name of an experimental activity is insufficient to communicate these nuances.

Box 1 presents several theoretical perspectives that extend our considerations about the effects of different activity characteristics on flow states. They propose that flow states vary when their challenges differ in quantity, certainty, potential, clarity, and subjective interpretability. A targeted empirical exploration of whether these different challenges lead to a uniform flow state or multiple variants with overlapping and unique characteristics remains underexplored. However, there are examples of flow being measured under different conditions that align with the framework presented in Fig. 1 and could inspire more targeted empirical approaches.

A common example of manipulating activity autonomy characteristics in flow studies includes those using arcade games like Tetris (red area in Fig. 1). These experiments typically create three conditions that demand different interaction speeds to control falling blocks (e.g.,[10,64,65]). Future research could investigate how interaction speed manipulation relates to the Integrated Model of Flow and Clutch States (see Box 1), as the enforcement of pressurised interaction speeds and fixed goals in these studies aligns more closely with the definitions of clutch than flow[66].

To our knowledge, no neurophysiological flow studies have yet explicitly manipulated strategy, goals, or feedback, though some have included features that may inspire future research designs. For example, Wolf et al.[67] asked participants to imagine table-tennis shot responses based on video stimuli in their electroencephalography (EEG) study—a paradigm that, if compared to actual play, might offer insights into challenges stemming from feedback manipulations. Similarly, dual-task studies incorporating auditory and visual distractor tasks alongside primary FPS gaming tasks (visual[37,68,69] auditory[40,70,71]) hint at potential challenge manipulations through goal quantity, which could test the Informational Theory of Flow[72,73] (see Box 1). Musical performance studies that contrast improvisation with fixed sheet music performance tasks manipulate the spectrum between internal (improvisation) and external (sheet music) feedback[74,75].

Although recent neurophysiological flow studies adopted these musical improvisation and sheet reading tasks separately[20,21,76], none have compared them experimentally yet. Future research could systematically manipulate these four autonomy dimensions to categorise different challenge types and explore their effects on flow states.

Mapping the complex differences in task challenges and their critical role in the challenge skill balance represents an opportunity for advancing the understanding of flow states and their neurophysiological correlates. We argue that the types of challenges across activities are not uniform, even when skill-balanced, and thus may operationalise flow heterogeneously, potentially explaining the inconsistent findings across neurophysiological flow studies[4]. The presented activity-autonomy framework enables a systematic approach to test this by distinguishing challenge types. We

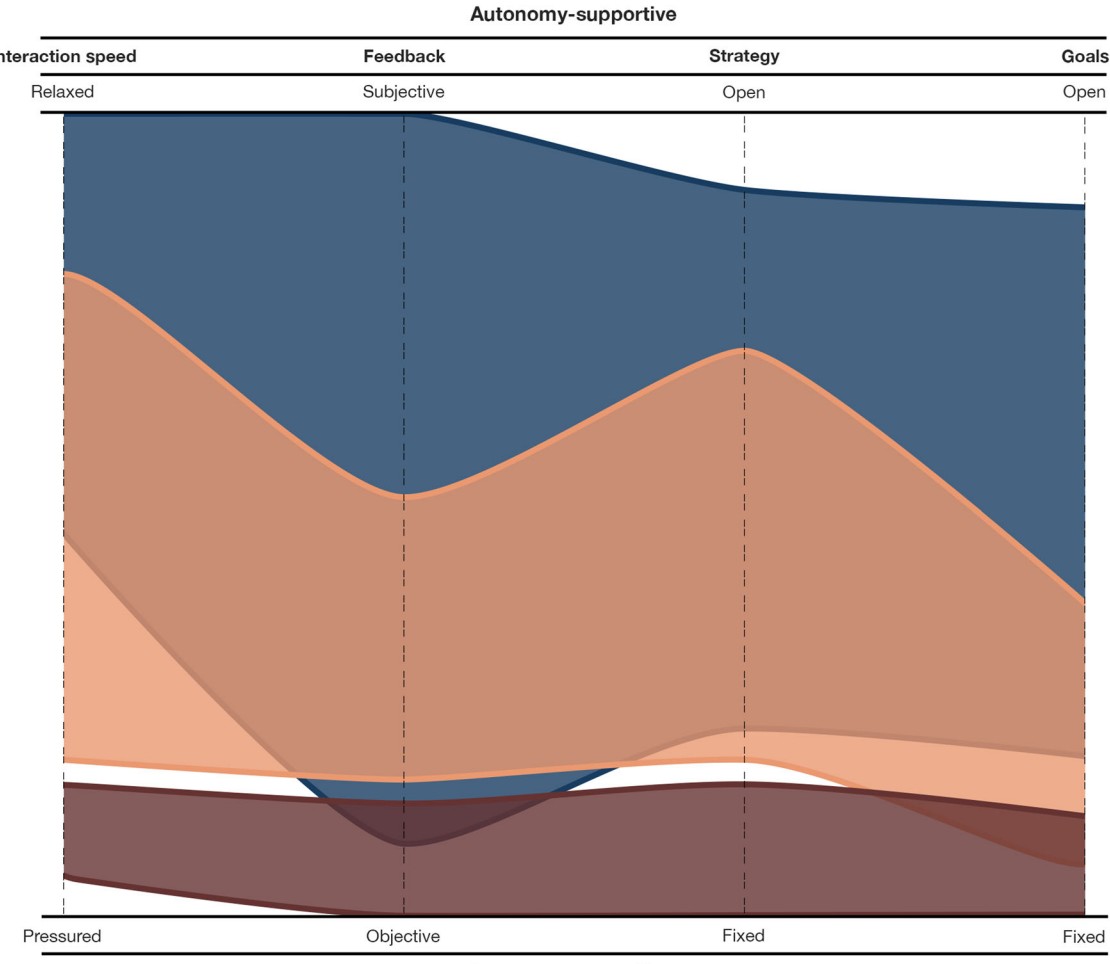

**Fig. 1 | Three examples of different activity-autonomy structures.** Red = Tetris, orange = rock climbing, blue = abstract painting. Interaction speed, feedback, strategy, and goals are autonomy spectrums that can change between activities and their contexts. Coverage towards the top represents autonomy-supportive activity characteristics and coverage towards the bottom represents autonomy-controlling activity characteristics. The coverage of activities on this graph is based on our subjective interpretation of the example activities. Researchers can use this diagram to identify where on these spectrums their experimental activities lie, and how they may share similarities or differences with other experimental activities. Tetris[36,122], abstract painting[1,123–126], and rock climbing[1,127,128] are frequently featured in flow literature. In Tetris, various iterations of four-square shapes fall sequentially from the top of a digital display, gradually pilling up at an increasing pace (objective, game-based feedback), and players must create horizontal lines from blocks to clear them (fixed goal) before block piles reach the top of the screen (pressured interaction speed). To achieve this goal, players can move blocks left or right or rotate them 90 degrees as they fall (fixed strategy) to fit them into gaps that make horizontal lines

(fixed goal). Unlike when making an abstract painting, there is no need or use in deviating from Tetris' prescribed rules or trying to make new goals, as this will inevitably lead to failure. Whilst making an abstract painting (blue area), if a painter imagines something pictorial they would like to paint or an emotion they strive to express (open goal), they can paint this however they like (open strategy), informing progress in the painting based on their aesthetic preferences (objective and subjective feedback) at any pace they like (relaxed interaction speed). During rock climbing, pre-set stages in a climb (fixed goals) can be approached by the climber in several ways (moderately open strategy). However, upon approaching pivotal challenges in the climb (objective feedback), the climber is faced with a fixed, unadaptable goal; they either climb it or they do not. Climbers with different levels of expertise may perceive subjective feedback to inform their actions differently than a novice, who is less informed on how to tackle such challenges. However, unlike making an abstract painting where the painter can freely change direction midway through the process, the climber must adhere to fixed goals enforced by the rockface.

## Box 1 | Four theories of different challenges types underlying flow states

**Informational Theory of Flow**[72,73]
This theory suggests several activity structural configurations predict flow state intensity (assuming the flow state is unidimensional). It proposes that flow intensity increases when there is an increased number of means (strategy) and/or ends (goals) in activities. In other words, if activity goals can be achieved via several strategies or if a single strategy could achieve several goals, the activity induces more flow than activities with single, fixed strategies and goals. The theory proposes that flow intensity increases with increased goal and/or strategy uncertainty. To resolve these uncertainties, a problem-solving process is prompted towards meeting desired outcomes. This problem-solving process is what triggers flow states.

**Challenges of the unknown**[1]
Csikszentmihalyi considered exploratory activities that present unknown challenges the most conducive for flow. He proposed activities can possess two types of unknown challenges; those that "lead to discovery, exploration, problem solution, and which is essential to activities like composing, dancing, climbing, and chess; or the most concrete challenge of competition, which is important in activities like basketball" ([1] p. 30). Others[129] have also suggested that activities with greater unknowns may house more opportunities for flow by way of resolving these unknowns.

**Automaticity and engaged mindedness**[130]
Waterman proposes how challenges are approached in different types of activities, which leads to different types of flow states. He suggests that flow in artistic creation activities (e.g., painting and music composition) is managed using internal cues, such as inspiration and intuition, which act as feedback to steer progress. In contrast, flow in pre-structured activities (e.g., athletics and arcade games) relies more on external cues, such as rules and scores to guide the activity progress. He distinguishes these activity types employing cognitive mechanisms differently (namely, system 1 and 2 configurations[131]). Whilst one can prepare for pre-structured contexts according to their rules, engaging in activities with unknown challenges lacks enforced rules and instead, relies heavily on momentary internal cues to make sense of how to engage with them. This perspective adds to the complexity of the immediate, unambiguous feedback antecedent since activities can involve different types of feedback; feedback that comes from the activity or from the self.

**Integrated Model of Flow and Clutch States**[66]
The recent conceptualisation of the clutch state developed as a branch of flow research, taking influence from Houge MacKenzie et al.'s[132] distinction between playful (paratelic) and serious (telic) flow states. In the model, both flow and clutch states are triggered by appraised challenges and share experiential characteristics such as absorption, automaticity, and enjoyment[66,133]. However, they are distinguished by how activity goals are subjectively interpreted. Swann and colleagues[133] suggest that while the flow state can be triggered by the exploration of open goals and feels effortless, the clutch state is triggered by the focused pursuit of achieving specific, fixed goals under pressure and feels effortful[133]. Clutch has only been studied in sports and adventure recreation contexts so far but recent developments, including theoretical frameworks[133], systematic reviews[134], qualitative studies[127,135–137] and self-report scales[66,138,139] have broadened our understanding of clutch states.

encourage its use in neurophysiological flow research to compare the outcomes of different activities and their unique challenges on flow states. Specifically, this could aid in the comparison of frequently tested autonomy-controlling activities with more autonomy-supportive activities, which remain largely overlooked.

## Methodological recommendations

Ensuring that flow states are genuinely induced in experiments relies on fulfilling necessary antecedent conditions, reducing extraneous effects, and analysing collected data systematically. However, the experimental designs of neurophysiological flow studies do not always meet these requirements. This can lead to inconsistencies in the states being measured between studies, despite their shared use of the term "flow". Box 2 presents a checklist of best practices in experimental methodology for these studies to mitigate these inconsistencies, grounded in the empirical evidence reviewed in this section.

### Participant skills and motivation

The emergence of flow depends on individuals possessing the necessary skills to achieve a balance between challenge and skill, as well as having self-efficacy and intrinsic motivation to engage in the activity or situation[2]. However, these prerequisites are sometimes overlooked in neurophysiological flow studies. In some studies, participants had no pre-established relationships with the task, either inadvertently[77] or intentionally by design[28,69,78]. Typically, participants are instructed by researchers to engage in activities rather than initiating the activities themselves. This formal instruction may diminish intrinsic motivation[51,79,80], introduce demand characteristics, and make participant engagement extrinsically motivated[25,81,82]. To overcome these challenges, some studies have adopted commendable practices to ensure necessary skill and motivation-relation conditions are fulfilled.

Several studies ensure that recruited participants possess the necessary skills and self-efficacy in their experimental tasks. One approach is to incorporate these characteristics into inclusion criteria, requiring participants to have a minimum duration of activity expertise or training[20,21,76,83]; a minimum frequency of weekly engagement in the activity[38,83,84]; active participation in activity-specific initiatives[85,86]; and/or past certifications, training, experiences, and/or achievements in the activity[21,64,76,87]. Additionally, standardised scales can be used to assess whether participants meet these inclusion criteria, particularly in domains with unique, domain-specific factors contributing to skills and expertise (e.g., in music[88] or videogames[89]). Another approach is to allow participants to practice experimental tasks in advance[78,87,90,91] and/or conduct pre-experiment performance tests to establish interindividual baseline competencies in the activity[65,84]. While there is no gold standard method for ensuring participants have the appropriate skills and self-efficacy to experience flow states in experimental tasks, future research may consider implementing these examples when designing inclusion criteria and screening procedures. Analyses could also include data on skills and expertise as covariates of flow outcomes.

Efforts aimed at increasing participants' intrinsic motivation to participate in research contexts include tailoring research designs to incorporate autonomy-supportive and personally incentivising features. This can involve allowing participants to choose tasks or stimuli beforehand to match personal interests[20,64,87]; using real-time human (instead of computer) interactions or engagements[64,85,91,92]; and/or choosing activities that the participants would be doing regardless of the research context[93–95]. While these approaches are expected to foster flow states by increasing intrinsic motivation, empirical evidence is lacking. Administering self-reports of intrinsic motivation in flow experiments, using instruments like the Situational Motivational Scale (SIMS)[96], can provide a direct measure of participants' motivational incentives for being part of flow research.

## Box 2 | Experimental flow research methodology checklist

**Instructions for use:** Researchers may review each checklist item adopted into their research design and mark with a 'x'. The citations next to each item are examples of past studies that have implemented the item principles. Some items offer alternative approaches to others on the checklist, meaning not all items can be adopted simultaneously. Additionally, some items may not be appropriate for all experiments. We recommend that future articles on the neurophysiology of flow may consider including the completed checklist in their supplementary materials.

| | | |
|---|---|---|
| **1** | **Terminology and reporting** | **(x)** |
| a | Explicitly acknowledge which necessary variables from theoretical frameworks are met and unmet in the study design | |
| **2** | **Experimentally manipulate challenge types independently or in combination** | |
| a | Task feedback (subjective to objective)[57,140] | |
| b | Interaction speeds (relaxed to pressured)[10,64,65] | |
| c | Strategies and goals (open to fixed)[90,102,141] | |
| d | Goals in co-occurring tasks (quantity)[37,40,68–71] | |
| e | Duration[40,99] | |
| **3** | **Personalise challenge levels to individual participants** | |
| a | Personalise participant-specific challenge levels based on a pre-experiment test performance[40,84,104] | |
| b | Personalise participant-specific challenge levels using a dynamic difficulty adjustment system[65,103] | |
| c | Make the relative differences in challenge equal between experimental conditions[40,84,104] | |
| **4** | **Ensure participants have adequate skills and self-efficacy in the study task(s)** | |
| a | Set thresholds on the duration of expertise or training in the activity[20,21,76,83] | |
| b | Set thresholds on the frequency of engagement in the activity (e.g., per week or month)[83,84] | |
| c | Confirm current active involvement in the activity[85,86] | |
| d | Confirm past certifications, training, experiences, and/or achievements in the activity[21,64,76,87] | |
| e | Use standardised scales designed to measure domain-specific expertise[20] | |
| f | Conduct a pre-experiment skills test to establish baseline competency in the activity[65,84] | |
| g | Allow participants to practice the task/stimuli beforehand to eliminate novelty effects[78,87,90,91] | |
| **5** | **Ensure participants are intrinsically motivated to do the study task(s)** | |
| a | Give participants a choice of experimental tasks/stimuli to match personal interest[20,64,86,87] | |
| b | Use real-time human interactions or engagement[64,85,91,92] | |
| c | Confirm that the activity has relevance, contributes to participants' personal motives, and/or is something they would engage in regardless of research participation[93–95] | |
| d | Administer pre-experimental self-report measures of situational motivation | |
| **6** | **Reduce individual difference effects between participants** | |
| a | Utilise within-participants designs where participants are tested in all experimental conditions[10] | |
| b | Utilise multi-task designs where participants are tested in all tasks[10] | |
| **7** | **Use and process self-report data appropriately** | |
| a | Remove cases where planned antecedent manipulations misalign with antecedent self-reports[116] | |
| b | Report and analyse isolated flow dimension self-report scores[28,113–115] | |

While intrinsic motivation may not be a necessary antecedent for all experimental research, it is necessary for flow according to recent empirically informed frameworks[2,7]. So, in addition to designing experiments that promote intrinsic motivation, it seems reasonable to directly ask participants about their motivational incentives for taking part in flow research.

### Procedural duration

Little attention has been paid to the time required for flow states to emerge once an activity has started. However, some studies have raised concerns about the short durations typically observed in neurophysiological flow experiments[4,41,97,98]. For example, Khoshnoud et al.[41] proposed neurophysiological flow studies using video game activities that last an average of 12 min. Limited findings suggest that even participants highly familiar with an experimental activity require a longer duration for flow state onset than this[41]. Yun et al.[40] tested expert video gamers playing a FPS game over one hour and gamers reported taking at least 25 min to enter flow states. This study showed participants a video replay of their session post-experiment and asked them whether they were in flow or not in every 5 min interval of the replay. While this approach offers some insight, albeit retrospectively, about the experienced flow states, it may also lead to heterogeneous interpretations of flow between participants. The temporal experience tracing self-report methods developed by Jachs et al.[99] represent a more complex version of this method and might be considered to measure isolated flow dimensions in a more standardised way. Of note, certain activities, such as FPS gaming, may represent clutch states due to their pressurised and fixed goal characteristics, which may require different durations for flow states to emerge compared to more autonomous activities. Similarly, de Manzano et al.[86] showed a gradual increase in physiological correlates of flow according to the flow state scale[100] over five trials of a 90-120 min piano playing study. However, since this was a correlational design, the physiological changes observed may reflect other experiential aspects of playing music repetitively. Future neurophysiological research should explore temporal changes during the flow state and the necessary duration for determining the onset of the flow state across different activities and participants.

### Personalising challenge levels

It is crucial to consider challenge levels in relation to individual participant skills when adopting the challenge skill balance paradigm in flow experiments. We have highlighted that this is the most prevalent experimental manipulation used in neurophysiological flow research and suggested that different types of challenges should be manipulated in isolation. Here, we show there is heterogeneity in how different studies set up challenge levels, which likely leads to participants experiencing flow in experimental conditions where it is not intended.

For researchers creating experimental conditions with different challenge-skill ratios, we advocate for calculating these ratios around each participant's skill level. Task challenges should be manipulated in all conditions, not just the condition thought to induce flow states. This approach ensures that conditions labelled as easy, optimal, and hard are genuinely experienced as such by each participant. This recommendation responds to some studies that use fixed challenge levels without tailoring them to individual participant skills, either in all[71,101] (Fig. 2a) or just in the easy and hard (non-flow) conditions[29,37,102] (Fig. 2b). With these approaches, a skilled participant (e.g., Participant 2 in Fig. 2) may experience flow in a hard condition, while an unskilled participant (e.g., Participant 1 in Fig. 2) may experience flow in an easy condition. These discrepancies can lead to significant variations in collected data, decreasing the chances of identifying consistent patterns between flow and non-flow conditions. To mitigate these problems, challenges should be made optimal for every participant in the flow condition, and the relative difference in challenge levels in non-flow conditions should represent standardised increases or decreases from the participants' baseline (Fig. 2c, d). This approach mitigates confounding effects caused by individual differences in skill levels.

There are two primary ways to achieve this. Harmat et al.[65] achieved this by having Tetris respond adaptively to real-time participant performance, where the hard experimental condition had a three-step speed

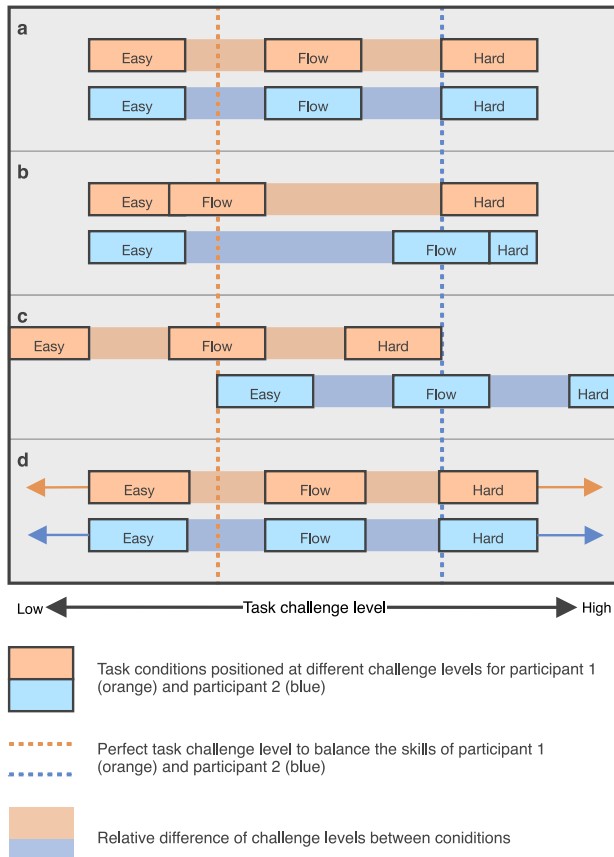

**Fig. 2 | Four approaches to designing experimental challenge skill balance conditions.** This figure shows how different experimental designs match task challenges with the skills of two hypothetical participants. The perfect challenge skill balance is lower for participant 1 (orange) than for participant 2 (blue), leading to variable adjustments in challenge levels between the four examples. **a.** Challenge levels in all conditions are fixed and not tailored to participants' skills. For participant 2, the flow condition is below their challenge skill balance, and for participant 1, the flow condition is above it. **b.** Challenge levels in easy and hard conditions are fixed but the flow condition is personalised to each participants' skills (e.g., by setting participants' task challenge level using their pre-experimental performance). Whilst flow is likely for both participants (since the flow condition is personalised), participant 2 may experience flow in the hard condition because the hard challenge level is nearly balanced with their skills (as indicated by the blue dashed line position). **c.** Challenge levels in all conditions are personalised to participant skills (e.g., using pre-experiment performance data). First, the flow condition challenge level is set, then, challenge levels in the easy and hard conditions are set based on a standardised decrease or increase (respectively) from the challenge level of the flow condition. This ensures the difference in challenge levels between all three conditions is consistent for every participant, unlike in a and b. **d.** Challenge levels in all conditions are continuously tailored to participant skills using real-time performance data. Challenge level in the flow condition is dynamically adjusted to real-time performance results and challenge levels in the easy and hard conditions are also dynamically adjusted at a standardised decrease or increase (respectively) from the flow condition difficulty. Like in c, this ensures the difference in challenge levels between all three conditions is consistent for every participant, unlike in a and b.

increase and the easy condition had a three-step decrease in block-fall speed from the participant baseline (their baseline represented a personalised challenge skill balance). A systematic review by Mortazavi et al.[103] demonstrates 85 other studies that use this dynamic difficulty adjustment method in gaming studies (not all flow studies). This method is illustrated in Fig. 2d. Other flow studies calculated the baseline and standardised challenge increase or decrease in other experimental conditions using the results of a pre-experimental screening procedure that measures individual performance[40,84,104]. This method is illustrated in Fig. 2c. Of note,

Joessel et al.[84] developed a commendable, multi-stage method to screen for individual skills in FPS games. These two methods ensure every participant experiences personalised challenge levels with standardised differences in the challenge levels between experimental conditions.

### Within-participant multi-task designs

We have argued that the characteristics of activities and participants may underlie differences in flow state outcomes. This may explain why single studies find significant neurophysiological markers of flow but these outcomes cannot be replicated in other studies that use different activities and participants[4]. Therefore, flow experiments must ensure activity and participant variability are controlled for. To mitigate potentially extraneous between-participant and between-activity effects on flow states, we recommend experiments that test participants in several tasks in all experimental conditions. Within-participant designs account for the confounding effects of individual differences because experimental condition outcomes can be compared at the individual level. Equally, these confounding effects are further controlled for when the same participants do several tasks in the same experiment (e.g., different iterations of the same task that differ in challenge types). This approach advances current practices of searching for patterns in the results of different experiments that use different tasks and different participants, which likely include confounding effects that mask information about flow states.

There are very few studies using multi-task, within-participant designs to research flow neurophysiologically. We commend these studies, however, note their results should be interpreted as preliminary at this stage due to having small sample sizes[105,106], and not personalising challenge levels to participants in all experimental conditions[70]. A notable result from a study using a multi-task within-participant design was found by de Sampaio Barros et al.[10]. Heart rate variability in four experimental conditions presented consistent patterns amongst participants within tasks (Tetris and Pong, two arcade games) but these consistent patterns were different between the tasks. Because the same participants did all experimental conditions in both tasks, the potential for individual differences to have erroneously caused this effect can be ruled out. This suggests that interaction speed challenges and flow states may cooperate heterogeneously between tasks; even considerably similar tasks.

Important assumptions should be considered when conducting within-participant, multi-task experiments. Unless researchers are studying order effects specifically, it is important to counterbalance or randomise the order of experimental conditions to avoid extraneous learning effects being carried from one condition to another. Sample size should also be considered since whilst power is increased in within- relative to between-participant designs (requiring fewer participants), contrasts and comparisons may need conservative corrections (e.g., Bonferroni or Helm) to mitigate Type-II errors. Snijdewint and Scheepers[107] show commendable consideration for this by conducting sensitivity analyses to detect required sample sizes for each hypothesis they tested and providing G*Power[108] logs in supplementary material. When these assumptions are accounted for, multi-task within-participant designs increase the reliability of identifying generalisable flow state findings.

### Using self-report flow scales

A common and commendable practice in neurophysiological flow research is to capture self-report data using standardised scale instruments alongside neurophysiological data capture. These instruments measure subjective experiences and can give meaning to co-occurring neurophysiological activity when analysed together. When doing so, standardised approaches to computing raw scale data into representative flow or flow dimension coefficients are crucial. Yet, as shown in Fig. 3, the way flow state scales are processed often violates this principle.

In many neurophysiological flow studies that analysed self-report with neurophysiological data, composite global flow scores are calculated and then regressed against corresponding neurophysiological data segments (e.g.,[20,67,87]; Fig. 3b). This is done to isolate the signals that occurred when

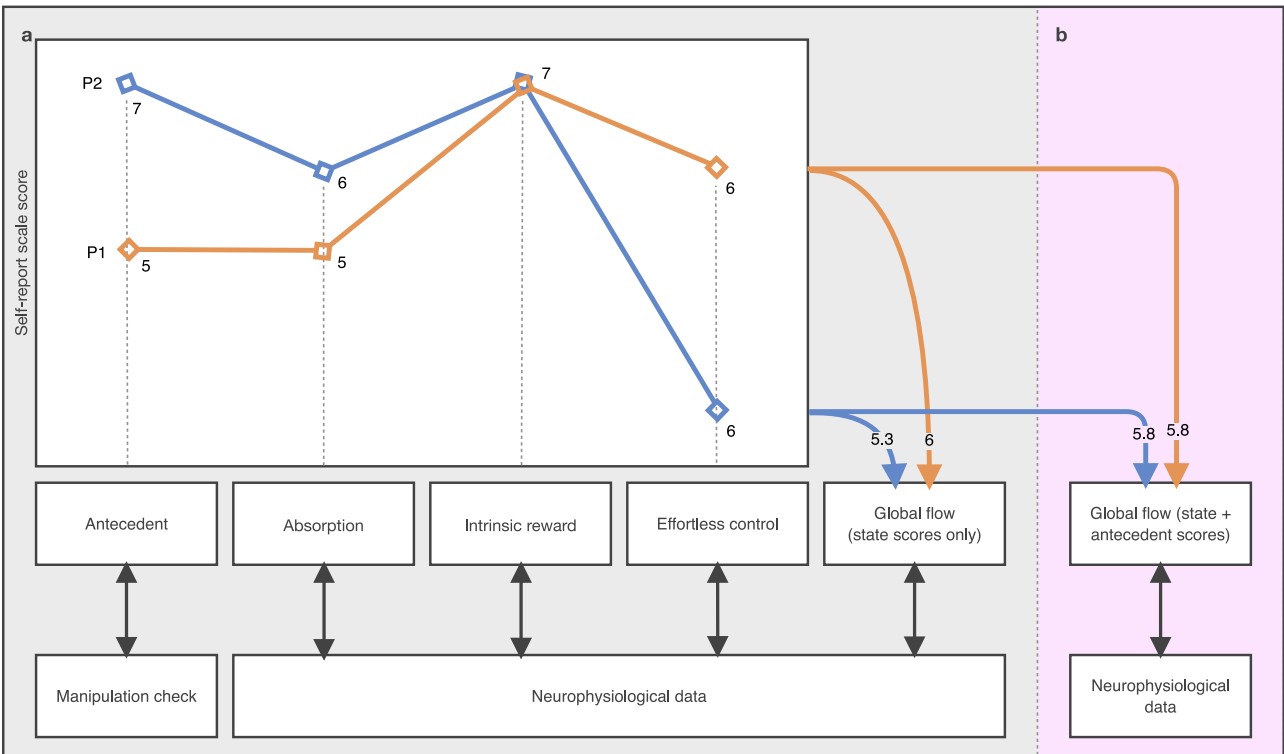

**Fig. 3 | Approaches to using and computing flow state self-report data.** P1 (orange) = participant 1 and P2 (blue) = participant 2. **a** P1 and P2 give different scores on the same flow dimensions. Antecedent scores are used to ensure experimental manipulations of antecedents correspond with self-reports, which validates flow was induced as designed. Flow state scores are used to run isolated and global analyses of neurophysiological data. Global flow scores are calculated by averaging state items only. **b** Global flow scores are calculated by averaging state and antecedent scores, unlike in a, which uses state scores only. The approach in b produces identical values for P1 and P2 (P1 = (5 + 5 + 7 + 6)/4 = 5.8, P2 = (7 + 6 + 7 + 3)/4 = 5.8), which differs from the approach in a (P1 = (5 + 5 + 7)/3 = 5.3, P2 = (7 + 6 + 7)/3 = 6). Calculations of global flow scores should consider excluding antecedents.

flow was reported, in search of the neurophysiological correlates of flow states. The composite scores are often calculated by averaging all the scale items, which follows the assumption that every item contributes to a single flow factor. Some researchers support this view, explicitly stating that flow is unidimensional[50,109,110] and others infer it using singular terminology ("flow"). However, whilst calculating global composite flow scores creates a convenient value to regress against other data, it discards evidence demonstrating the nuanced multi-dimensional structure of flow. For example, Norsworthy et al.[2] propose flow states consist of three dimensions: absorption, effortless control, and intrinsic reward. In different combinations, these dimensions may represent several unique state variants.

The convention of averaging self-report data to generate global composite flow scores prevents unique flow state variants from being discovered. For example, in Fig. 3, participants 1 (orange) and 2 (blue) report variable scores between flow dimensions. However, when computed into a composite score, their scores are identical. Additionally, studies using factor analysis show that different flow dimensions account for different amounts of variance in the overall flow state[111,112]. For example, the psychological flow scale (PFS[111]) showed that absorption, intrinsic reward, and effortless control dimensions explained 53.29%, 51.84%, and 28.09% of the overall flow state respectively (these values are based on our calculations of published data). Consequently, averaging scores across dimensions without considering their relative or isolated importance may not provide an accurate representation of a participant's flow state (Fig. 3b). Researchers should include analyses at the dimension level, as well as the global level when using self-report data to make predictions about neurophysiological data (Fig. 3a).

The calculation of global composite flow scores also commonly contains data unrelated to the flow state. Flow antecedent information (e.g., challenge skill balance, clear goals, and unambiguous feedback) are

frequently included in these coefficient calculations (with exceptions from[28,76,113–115], see Fig. 3b). Yet, many agree that antecedent items provide information about whether theoretical requirements to achieve flow have been met, which is distinct from the flow state experience (e.g.,[111,112]). We recommend using antecedent self-report data to check whether corresponding experimental manipulations were experienced as expected (Fig. 3a). For example, to check whether the task was reported as optimally challenging when it was designed to be (e.g.,[101,116,117]). We recommend researchers report how these principles are applied—or not—to facilitate a deeper, transferable understanding of flow state neurophysiology between studies.

## Outlook

Flow is a unique state associated with optimal performance, and is robustly linked with mental health, flourishing, and well-being[118–120]. Recent research also suggests a causal protective effect of flow disposition against physical health problems[121]. Unsurprisingly, there has been intense interest in revealing the neural correlates of flow; yet, the findings have been inconsistent and a clear neurophysiological mechanism remains elusive. In this article, we have identified and discussed methodological reasons behind these inconsistencies. We propose refined methodological approaches for conducting flow state research with neurophysiological measures, intending to improve the consistency, quality, and scope of such research. Addressing these methodological issues is critical for tackling the replication problem identified in neurophysiological flow studies.

We advocate for a systematic exploration of various types of challenges and their potential diverse effects on flow state outcomes. We address this issue by (i) introducing a testable activity-autonomy framework with four levels of autonomy that can vary within activities (such as interaction speed, feedback, strategy, and goals; Fig. 1); (ii) reviewing theoretical perspectives

**Perspective**

on different types of challenges associated with flow states; and (iii) offering concrete suggestions for experimenting with new challenge types. Further exploration into the challenge skill balance and other important flow antecedents could reveal the reasons behind the heterogeneous neurophysiological findings related to flow in past studies.

Future research in flow state neurophysiology is further encouraged to ensure that all necessary requirements for achieving flow are fulfilled. To this end, we have prepared a checklist to guide researchers. We also discuss strategies for recruiting and motivating skilled participants for complex, within-participant experiments, the need to explore the effects of procedural duration and advocate for a standardised analytical approach to processing self-report data, crucial for interpreting neural data. The proposed checklist consolidates these recommendations, serving as a guide for future flow studies, not limited to neurophysiological studies. Through the systematic application of these refined methodologies, we aspire to catalyse future research on flow that not only reveals the intricate neurophysiological basis of flow states but also fosters a more nuanced appreciation of their complexity and variability across different contexts and domains. This considered endeavour may hold the promise of bridging the gaps in our understanding, paving the way for innovative applications that enhance human potential, flourishing and well-being on a broader scale.

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

## Acknowledgements
O.D. would like to thank Alan Waterman for his discussions about flow theory whilst writing this article.

## Author contributions
O.D. conceptualised and wrote the original draft and produced all the figures. P.H. and J.B. provided joint research supervision for this work.

## Competing interests
The authors declare no competing interests.
