## [Peer Review File · Communications Psychology]

14th Dec 23

Dear Oliver,

Thank you for your patience during the peer-review process. Your manuscript titled "In search of authentic flow: Challenges in neurophysiological studies" has now been seen by 2 reviewers, and I include their comments at the end of this message.

The reviewers are on balance enthusiastic about your work. However, they also mention several significant concerns. We are interested in the possibility of publishing your manuscript in *Communications Psychology*, but would like to consider your response to these concerns in the form of a revised manuscript before we decide on publication.

In detail, the referees, and in particular Reviewer #2 highlight that there are multiple instances throughout the paper where either the issue you identify in the field, or the solutions you propose, are not compellingly supported by references to the literature.

Any description of existing issues must be persuasively demonstrated based on existing empirical evidence. Conversely, because the nature of your submission is a Perspective, it's permissible for some suggestions as to how existing issues can be overcome to be speculative (with their hypothetical suitability flagged). However, this must be strictly limited to a few instances, where you lay out how their success may be established. In general, the description of solutions should be specific and detailed, to maximize their utility to other researchers in the field.

In sum, we invite you to revise your manuscript taking into account all reviewer and editor comments.

EDITORIAL POLICIES AND FORMATTING

You will find a complete list of formatting requirements following this link:
<https://www.nature.com/documents/commsj-style-formatting-checklist-review-perspective.pdf>

Please use the checklist to prepare your manuscript for resubmission.

* TRANSPARENT PEER REVIEW: Communications Psychology uses a transparent peer review system. This means that we publish the editorial decision letters including Reviewers' comments to the authors and the author rebuttal letters online as a supplementary peer review file. We publish these records for all accepted manuscripts. However, on author request, confidential information and data can be removed from the published reviewer reports and rebuttal letters prior to publication. If your manuscript has been previously reviewed at another journal, those Reviewers' comments would not form part of the published peer review file.

If you have any questions about any of our policies or formatting, please don't hesitate to contact me.

Please use the following link to submit your revised manuscript and a point-by-point response to the referees' comments (which should be in a separate document to any cover letter):

[link redacted]

We hope to receive your revised paper within 12 weeks; please let us know if you aren't able to submit it within this time so that we can discuss how best to proceed. If we don't hear from you, and the revision process takes significantly longer, we may close your file.

We understand that due to the current global situation, the time required for revision may be longer than usual. We would appreciate it if you could keep us informed about an estimated timescale for resubmission, to facilitate our planning. Of course, if you are unable to estimate, we are happy to accommodate necessary extensions nevertheless.

Please do not hesitate to contact me if you have any questions or would like to discuss these revisions further. We look forward to seeing the revised manuscript and thank you for the opportunity to review your work.

Best wishes,

Marike

Marika Schiffer, PhD

Chief Editor

Communications Psychology

REVIEWERS' EXPERTISE:

Reviewer #1 flow state, cognitive neuroscience

Reviewer #2 flow state, organisational psychology

REVIEWERS' COMMENTS:

Reviewer #1 (Remarks to the Author):

Basic reporting

This manuscript is a thoughtful review on the wide range of assumptions underlying the classic experimental approaches to investigate flow state. The authors argue that research in this field, especially those studies aimed at unravelling the neurophysiological mechanisms of Flow, has systematically neglected some key factors (e.g., autonomy within activities, participant characteristics, duration of experimental situations, etc.). These factors may potentially modulate the Flow state, which could explain the inconsistent results that have been found so far. Furthermore, they point out the interesting idea that Flow should not be considered as a unidimensional state and stress the relevance of differentiating it from other states such as clutch.

In my opinion, this is a well-written manuscript tackling a significant and timely topic. The ideas are well organized and well argued. I really enjoyed the summary of the state of this area of research and consider it a very good contribution for the potential readers of the journal. I think that, essentially, the manuscript could be accepted for publication as it stands except for a couple of typos.

As far as the content itself is concerned, I have a few issues I'm going to raise, as contributions aimed at possible improvements to the manuscript. Below I do detail also some errors and more specific aspects.

The title of the manuscript does not, in my opinion, reflect the content of the paper. The word authentic is perhaps the most controversial since the paper does not discuss the existence or not of Flow but rather its possible variants. I think that removing this word from the title might be a good idea.

In the first paragraph of the introduction, the authors mention different situations in which the emergence of the Flow state has been reported (i.e., climbing, writing, and dancing). They go on to point out that these are activities that involve intrinsic motivation to engage in challenging situations. However, I believe that in the case of dancing (perhaps also in writing although to a lesser extent), it cannot be considered as a challenging activity, especially in free dance contexts. I would clarify this issue.

On page 6 the issue of whether the Flow state is necessarily an effortless state or not is discussed. That is, can Flow exist in activities that involve great physical or mental effort? I think this is a very interesting aspect that the authors could delve into in order to give an explanation within their framework.

In a similar vein, at the end of page 6, the authors mention "the dark side" of Flow. I am referring to the relationship between Flow and gambling problems, addictions, or disorders such as anxiety or depression. I wonder in what sense the framework proposed could explain these seemingly unrelated situations.

On page 9, the authors suggest the idea that chess or basketball are not improvisational activities like dance or music. While it is true that in chess and basketball there are well defined goals (winning), they are activities with a very high improvisation component, sometimes perhaps even greater than in dance or music where choreographies or musical pieces are usually performed, or even improvised but based on previously learned structures. On the other hand, basketball, like chess, involves a high creative component in each move. In this line, on page 16 it is stated that painting allows for exploration and open-ended objectives. However, it would be good to clarify that this only applies to certain types/styles of painting since some styles (e.g., hyperrealism or portraiture) do have concrete objectives and do not allow for exploration as such. The authors could clarify what they mean in these cases.

Also on page 9, statistical data is given which in my opinion is not necessary in this type of article:

“...(Fishers $z = 0.73$ from correlations of 11 studies) or education contexts (Fishers $z = 0.32$ from correlations of 16 studies)”

A key aspect that could be improved in the manuscript concerns the authors' methodological proposal. While some experimental design ideas for future studies are mentioned throughout the text and even a collective effort to experimentally test possible variants of Flow is urged (page 16), I missed a specific section with concrete recommendations including some examples of experimental designs that would allow future researchers to follow a concrete line. An example of this can be found in the exercise-cognition field (Basso and Suzuki, 2017).

Also, on page 16 there is a sentence that could be controversial:

“These provide the appropriate tools to identify the state’s unique neurophysiological signatures and distinguish them from those of flow in a wide range of activities. “

Perhaps it is too risky to think, with the evidence to date, that there are particular neural signatures of Flow state.

In the first paragraph on page 19 there is a typo and should be rephrased for clarity.:

“However, since this was a correlational research design, the physiological changes may just reflect other experiential aspects of playing music repetitively. despite neurophysiological flow studies often being particularly short.”

I hope that this review will provide clear and constructive comments and suggestions that will help the authors to improve their manuscript so that it’s ready for acceptance.

Luis F. Ciria

REFs

Basso, J. C., & Suzuki, W. A. (2017). The effects of acute exercise on mood, cognition, neurophysiology, and neurochemical pathways: A review. *Brain Plasticity*, 2(2), 127-152.

Reviewer #2 (Remarks to the Author):

In general, the paper is written okay and the topic of the paper was interesting. I also wish to praise the authors for the literature review. It is also true that many of the things that were raised in the paper make sense. Having said this, however, I must admit I did not see a clear contribution of the paper. The authors seem to analyze experimental flow research and then provide a range of critique and recommendations. But, overall, the critique seem too general to be really useful. Moreover, the authors do not seem to provide clear and realistic recommendations for how to address the issues that researchers encounter when trying to study flow in the lab.

For example, the authors recommend that researchers use tasks that are really flow-inducing. Who would disagree with that? Of course, it is very likely that, when designing their studies, researchers do indeed spend quite some effort trying to come up with tasks that can be used to induce and measure flow. Yet, such endeavor is often constrained by the settings of the design and measurements. For example, I can imagine that scholars who include EEG would prefer to use a highly interesting, motivating, and complex task (e.g. game). However, when interested in specific EEG components, that is simply not possible. In such a context, a standardized task is needed in which it is clear when trial begins, when the stimulus appears etc. and that given everything else equal.

Saying to scholars who are struggling with such issues that they should use a better task, but without providing solutions to the specific problems would not be very helpful. It feels a bit is like doing one's best to repair something, while someone stands on the side, saying that you do it all wrong but without giving useful advice (that is not autobiographic, by the way ;0))

In the last part of the paper, the authors provide a summary of the points they raised and the specific points they challenged. This summary, however, also shows the limitations of the critique. Specifically, the authors state that their goals were to challenge the notion that flow...:

1) ... is a unidimensional state.

From my knowledge of the literature, I never encountered an article/chapter in which a scholar considered flow as a unidimensional state. Each researcher acknowledges that flow is a complex state with multiple facets.

2)... Can be operationalized via challenge-skill balance only.

Although it is true that flow many studies have used this approach, it is not clear whether researchers truly generally assume that this is the only method. Moreover, in a paper that wants to address this topic, one would expect some very specific recommendations, but these do not seem to be present in this paper. Merely saying that there are other ways to measure something would not suffice in order to provide a useful contribution to the literature.

3)... Cannot be experienced by activity-naïve individuals.

It makes perfect sense that it is probably easier to find flow in people engaging in their favored and optimal activities. For example, chess players, dancers, scientists doing science, etc. I can't imagine there would be many researchers who disagree with that statement. But in this paper, there are several sides to that. First, the authors do not provide any convincing discussion of previous empirical studies showing that flow cannot be induced in participants in a lab setting. They just used the arguments that the original conceptualization of flow was based on observing experts.

What evidence do the authors have (other than anecdotal) that flow cannot be induced in the lab? Do the authors think that this would only apply to flow or do they consider validity a general issue with experimental research?

4)... can occur over shorter periods

The authors mention the findings of two studies. In one study with a game, flow brain activity seemed to occur after 25 minutes (but how about the experience of flow?). In the other study, they refer to a study with piano players in which flow indicators seemed to occur after 90 minutes. In what way is that convincing? Would the authors maintain the view that, in principle, it is impossible (or very unlikely) that a skilled piano player can play a short song (say somewhere between 5 and 1 minutes) while in a flow.

I can probably mention several more points, like the ones above. But they would only be variations of the same theme. So, my overall idea of the paper is:

I like the topic, but I have the impression that the authors present much points of critique without strong empirical or conceptual evidence to back up their claims. Moreover, much of the points of critique that are provided seem to be points that, as far as I can judge, most scholars would be aware of but are often simply very difficult to deal with. Saying that researchers need better tasks, better participants, and better flow measures would provide valid points if the authors accompany that with useful advice on the real issues that flow researchers are struggling with.

I am sorry I cannot be more positive about this paper. My intention was not to be overly harsh, and I appreciate that the authors addressed this topic. I actually also agree that it would be useful to discuss several of the issues raised in the paper. However, above I just provide my view on the present paper.

Response to Referees

Summary of responses from authors:

We thank both reviewers for their time and careful consideration of the manuscript. Their insightful questions and suggestions have played a crucial role in enhancing the overall quality of this revised draft. We have provided individual responses to each point made in this document and signposted the corresponding line numbers in the manuscript. Here is the list of major changes made in the revision:

- (i) We have added a new section, “Recommendations for experimental flow research methodology” (522-773). This section synthesises insights from previous discussions, clearly outlines methodological challenges, and provides extensive and diverse solutions to address these issues.
- (ii) To further aid in the application of our recommendations, we have included a checklist that succinctly lists all the recommendations made for accessible use when designing research (503-521). Additionally, we have created a supplementary interactive PDF that allows researchers to document their use of the recommendations and include it as supplementary material in their published research.

In addition to these major changes, we have made updates across various sections of the article in light of the reviewer’s feedback to improve the clarity, coherence and flow of our arguments. Accordingly, we have updated the abstract (2-15), section titles, and the introductory and summary paragraphs of most sections to ensure they prime and remind the reader about section contents (251-268, 392-408, 493-499, 522-527). We integrated the previous section “Flow and clutch...” into the theoretical perspectives section (452-464), now titled “Types of challenges underlying flow states”. Moreover, the content from “Research design in flow experiments” has been expanded to form the basis of the new section, “Recommendations...” (522-773). Finally, we have rewritten the Concluding section to offer a forward-looking perspective (774-end).

RESPONSES TO REVIEWERS' COMMENTS:

Reviewer #1 (Remarks to the Author):

1. This manuscript is a thoughtful review on the wide range of assumptions underlying the classic experimental approaches to investigate flow state. The authors argue that research in this field, especially those studies aimed at unravelling the neurophysiological mechanisms of Flow, has systematically neglected some key factors (e.g., autonomy within activities, participant characteristics, duration of experimental situations, etc.). These factors may potentially modulate the Flow state, which could explain the inconsistent results that have been found so far. Furthermore, they point out the interesting idea that Flow should not be considered as a unidimensional state and stress the relevance of differentiating it from other states such as clutch.

In my opinion, this is a well-written manuscript tackling a significant and timely topic. The ideas are well organized and well argued. I really enjoyed the summary of the state of this area of research and consider it a very good contribution for the potential readers of the journal. I think that, essentially, the manuscript could be accepted for publication as it stands except for a couple of typos.

1. Thank you for your encouraging and predominantly positive feedback. We are confident that incorporating your comments has strengthened the revised version. Further, we have added new sections concerning your comments about effort, variability in goals and strategy within single activities, and solutions and recommendations for researchers. We expand on how we have done this in our proceeding responses to your comments.

As far as the content itself is concerned, I have a few issues I'm going to raise, as contributions aimed at possible improvements to the manuscript. Below I do detail also some errors and more specific aspects.

2. The title of the manuscript does not, in my opinion, reflect the content of the paper. The word authentic is perhaps the most controversial since the paper does not discuss the existence or not of Flow but rather its possible variants. I think that removing this word from the title might be a

good idea.

2. Thank you for this comment. We agree, and subsequently removed the term “authentic.”

3. In the first paragraph of the introduction, the authors mention different situations in which the emergence of the Flow state has been reported (i.e., climbing, writing, and dancing). They go on to point out that these are activities that involve intrinsic motivation to engage in challenging situations. However, I believe that in the case of dancing (perhaps also in writing although to a lesser extent), it cannot be considered as a challenging activity, especially in free dance contexts. I would clarify this issue.

3. Thank you for raising this important point about the variability in challenge levels across activities. Your point regarding the distinct nature of challenge in activities like free dancing and writing, which allow for a great deal of creative freedom, versus climbing, which typically involves more defined physical challenges, is insightful.

Because the Introduction, and especially its first paragraph, must introduce key concepts as clearly as possible, we have approached this issue with caution to avoid premature engagement with these complex conceptual problems so early. So, we have made the following changes:

(a) removed the term “challenging” from the first paragraph of the Introduction (27)

(b) added a few new sentences to question the uniformity of challenges across exploratory and creative activities compared to those with intense physical demands in the section: “flow state variability” (285-291).

(c) added further clarity at the start and end of paragraph 2 in the “autonomy structures within activities” section (348-349 and 383-391). These changes highlight the broad spectrum of autonomy that can exist within the same activities, depending on context, goals, and motivations.

Further, we have discussed the discrepancy in types of challenges extensively in the section “types of challenges underlying flow states”. (392-499)

We believe these changes address this important argument, which stems from a somewhat ambiguous understanding of what constitutes a challenge. We have delved into this issue later to ensure that key concepts are introduced clearly and in isolation beforehand.

4. On page 6 the issue of whether the Flow state is necessarily an effortless state or not is discussed. That is, can Flow exist in activities that involve great physical or mental effort? I think this is a very interesting aspect that the authors could delve into in order to give an explanation within their framework.

4. Thank you for raising this interesting issue. We interpret your comment about effortlessness as referring to variability in the objective challenges demanded by an activity. For example, an activity like motorcar racing demands great mental and physical effort, which raises the question of whether such high levels of effort might preclude the possibility of achieving a flow state. Although we are cautious about fully adopting this suggestion that this could offer explanatory support of the framework for the reasons below, we have nevertheless made some changes to reflect a nuanced view on this issue.

Primarily, our hesitancy stems from what we perceive as having a different interpretation of where “effort” comes into the flow framework. It is clear that different activities demand different levels of effort. However, our reference to effortlessness is less about the activity and more about the state emerging from engaging in the activity. We posit that a motorcar racing driver not in a flow state will perceive the activity as more effortful compared to one who is in a flow state. In other words, we view effortlessness as an indicator of the flow experience, rather than a trigger or antecedent for flow. Flow does not necessarily occur with every engagement in an activity; but when it does, the experience becomes more effortless.

Addressing your point about effortlessness in terms of the objective challenge level of activity, we believe that the concept of challenge-skill balance adequately assesses the

effort required by an activity. Nevertheless, to aid researchers in quantifying the objective challenge of tasks, we have included a new section, “Personalising challenge levels” in our new recommendations section (638-675). In Fig. 3 of this section, the light blue backgrounds represent objective challenges and the grey backgrounds represent the alignment of individual participants’ skills with the objective challenge spectrum.

Regarding how to measure the level of effort experienced once engaging in the activity, we find that state measures of effortlessness are sufficient for capturing the flow experience independent of the objective challenges imposed by the activity.

Furthermore, in the section, “The concept of flow” (first paragraph), we have included a discussion on the ongoing debate over the “level” of challenges (or effort) required to induce flow or its various intensities. Empirical evidence suggests that challenges in the challenge-skill balance must be “high” to facilitate more intense flow states, so we have cited Giovanni Moneta’s 2021 review article on this subject too. (130-132)

5. In a similar vein, at the end of page 6, the authors mention "the dark side" of Flow. I am referring to the relationship between Flow and gambling problems, addictions, or disorders such as anxiety or depression. I wonder in what sense the framework proposed could explain these seemingly unrelated situations.

5. We agree that the dark side of flow presents an intriguing subject matter, albeit one that is complex, vast and under-researched. However, this is beyond the scope of this perspective article. Of note, the primary focus of this article is on the state of flow rather than its outcomes.

6. On page 9, the authors suggest the idea that chess or basketball are not improvisational activities like dance or music. While it is true that in chess and basketball there are well defined goals (winning), they are activities with a very high improvisation component, sometimes perhaps even greater than in dance or music where choreographies or musical pieces are usually performed, or even improvised but based on previously learned structures. On the other hand,

basketball, like chess, involves a high creative component in each move. In this line, on page 16 it is stated that painting allows for exploration and open-ended objectives. However, it would be good to clarify that this only applies to certain types/styles of painting since some styles (e.g., hyperrealism or portraiture) do have concrete objectives and do not allow for exploration as such. The authors could clarify what they mean in these cases.

6. Many thanks for your insights, which resonate with our discussions regarding the level at which goals operationalise variation in flow. Your comments have reinforced our decision to include strategy and goals into our framework, highlighting the notion that different strategies can be employed to achieve the same goal, or conversely, the same strategies can be used to meet several different goals. This complexity underlines the importance of measuring these variables independently in research. Without such measurement, individuals participating in the same activity (e.g., basketball) might pursue different strategies and goals, leading to variability in psychological states that could otherwise remain unexplained by researchers.

Following your recommendation we have emphasized that activities can exhibit a wide range of autonomy characteristics (e.g., goals and strategies used in the same activity may vary greatly between contexts). This has been articulated in the first paragraph of “Flow state variability” section (272-277) and the second and last paragraphs of the “Autonomy structures within activities” section (348-350 and 383-391),

Further, after your recommendation, we have also advanced our exploration by advocating for experimental research to test the effects of different goal types. We have synergised our own ideas about how goals can be studied in the last part of “Type of challenges underlying flow states”, specifically by exploring different goal quantities (478-499). Furthermore, we have included the isolated manipulation of goals in the practical checklist based on these ideas and cited studies that could be developed to implement these concepts (515)

7. Also on page 9, statistical data is given which in my opinion is not necessary in this type of article:

“(Fishers $z = 0.73$ from correlations of 11 studies) or education contexts (Fishers $z = 0.32$ from correlations of 16 studies)”

7. These stats have now been removed (279-281).

8. A key aspect that could be improved in the manuscript concerns the authors' methodological proposal. While some experimental design ideas for future studies are mentioned throughout the text and even a collective effort to experimentally test possible variants of Flow is urged (page 16), I missed a specific section with concrete recommendations including some examples of experimental designs that would allow future researchers to follow a concrete line. An example of this can be found in the exercise-cognition field (Basso and Suzuki, 2017).

8. Thank you. We agree with this comment and have restructured the previously titled section, “Research design in flow experiments” to “Recommendations for experimental flow research methodology” as mentioned earlier in this document (522-777). This revamped section highlights limitations and offers practical solutions for each, drawing on past examples of best practices, and our own suggestions. We have integrated them with the previously proposed framework and included them in a checklist (501-521). This checklist is designed to guide researchers in refining their research designs.

9. Also, on page 16 there is a sentence that could be controversial:

“These provide the appropriate tools to identify the state’s unique neurophysiological signatures and distinguish them from those of flow in a wide range of activities. “

Perhaps it is too risky to think, with the evidence to date, that there are particular neural signatures of Flow state.

9. Thank you for your comment here, about our closing sentence in the “Flow and clutch: open and fixed goals” section of the manuscript (which has now been integrated into the “autonomy structures within activities” section (452-464)). We acknowledge your

suggestions and have subsequently moderated our assertion, considering the exploratory nature of research in this area in the last sentence of this paragraph.

10. In the first paragraph on page 19 there is a typo and should be rephrased for clarity.:

“However, since this was a correlational research design, the physiological changes may just reflect other experiential aspects of playing music repetitively. despite neurophysiological flow studies often being particularly short.”

10. Thank you. Removed.

Responses to Reviewer #2

1. In general, the paper is written okay and the topic of the paper was interesting. I also wish to praise the authors for the literature review. It is also true that many of the things that were raised in the paper make sense. Having said this, however, I must admit I did not see a clear contribution of the paper. The authors seem to analyze experimental flow research and then provide a range of critique and recommendations. But, overall, the critique seem too general to be really useful. Moreover, the authors do not seem to provide clear and realistic recommendations for how to address the issues that researchers encounter when trying to study flow in the lab.

1. Thank you for this insightful critique. We understand the concerns raised could be improved by proposing concrete, clear, realistic, and specific solutions to our critiques of the experiment research. Therefore, we have now added a new section, “Recommendations for experimental flow research methodology” (522-777) and a checklist “Experimental flow research methodology checklist” (501-521). Our intent is not to be overly critical but rather to contribute as active researchers in the field who are committed to the clarity and precision of what we and other researchers measure. We acknowledge that meeting all our recommendations may be challenging, but we hope this article will be of some assistance to the researchers in recognizing and articulating areas that need further investigation. Your critical comments have been instrumental in guiding these necessary changes. Thanks again.

2. For example, the authors recommend that researchers use tasks that are really flow-inducing. Who would disagree with that? Of course, it is very likely that, when designing their studies, researchers do indeed spend quite some effort trying to come up with tasks that can be used to induce and measure flow. Yet, such endeavor is often constrained by the settings of the design and measurements.

2. Thank you for your comments. We will first address your comment concerning recommending “tasks that are really flow-inducing”. We acknowledge a common oversight in most of the flow literature is that the “task” has been considered in isolation as either flow-inducing or not. Yet, this approach ignores the importance of the pre-established task/activity relationships of participants, potentially leading to inconsistencies in results. While it is a common understanding that tasks should be flow-inducing, the distinctive argument we present is that researchers must consider not only the tasks themselves but also the individuals involved if they want to achieve flow. The examination of participant-activity relationships has been largely overlooked, and this article aims to steer researchers towards incorporating these important components into their studies. In summary, we propose any task can induce flow provided it sufficiently challenges the participants’ skills, and the participants are intrinsically motivated to engage, which we have called the participant-activity relationship.

We have provided brief historical context and evidence for the importance of participant-activity relations in the first and second paragraphs of “Participant skills and motivation” (528-553, this section was previously titled “Participant inclusion criterion: motivation and skill level”). In the third and fourth paragraphs of this section, we have aggregated several approaches considered best practices from the limited studies adopting these principles (554-585). This section also includes our recommendation to use motivation self-report measures as a way to understand participants’ motivational stance in the experiment. Additionally, we have implemented these best practice examples into the checklist (points 3 and 4), enabling other researchers to readily apply them (516 and 517).

3. For example, I can imagine that scholars who include EEG would prefer to use a highly interesting, motivating, and complex task (e.g. game). However, when interested in specific EEG components, that is simply not possible. In such a context, a standardized task is needed in which it is clear when trial begins, when the stimulus appears etc. and that given everything else equal. Saying to scholars who are struggling with such issues that they should use a better task, but without providing solutions to the specific problems would not be very helpful. It feels a bit is like doing one's best to repair something, while someone stands on the side, saying that you do it all wrong but without giving useful advice (that is not autobiographic, by the way ;0)

3. Thank you for your comments emphasizing the importance of selecting tasks that are engaging and relevant to participants in studies aimed at measuring flow states, especially considering the challenges that have previously prevented this. We agree that our initial submission was insufficient in providing practical solutions. As mentioned earlier, we have now introduced a dedicated section for recommendations (522-777) that provides clear and detailed guidelines to help design experiments adopting interesting and motivating tasks. We have discussed several ways to ensure tasks remain interesting and motivating:

(a) We discussed pre-established task relationships in response to your last comment.

(b) In the “Procedural duration” section (586-609), we have highlighted methods that assess temporal changes in self-reports over the course of a task. Indeed, a task may be interesting and motivating 20 minutes in but not 5 minutes (or vice versa). We have reviewed studies identifying temporal changes in flow states and discussed the methodologies used to track these changes.

(c) In the new “Personalising challenge levels” section (638-675) and corresponding figure above this section (610-637), we draw attention to common mistakes in experimental design, particularly in establishing conditions along the challenge-skill balance spectrum. When challenges are not properly balanced with participants' skills, motivation or interest will wane, as is supported by previous research. This section and Fig.3 show both “imbalanced” (a and b in Fig.3) and “balanced” (c and d in Fig.3) conditions in

experiments, providing examples for each scenario and guiding the design of challenge-skill balance conditions. Our recommendations to implement these best practices have been incorporated into the methods checklist (501-521).

These additions address our concern that researchers may incorrectly claim to have measured a flow state without meeting all the necessary preconditions (e.g., lack of intrinsic motivation) or by setting up experimental conditions sub-optimally (e.g., an unintended imbalance between challenges and skills). Such issues complicate the interpretation of the already complex flow literature. While we acknowledge the need for well-controlled research designs, it is critical to offer guidelines and recommendations for satisfying these necessary conditions optimally. Where research falls short of these preconditions, we argue that claims of inducing and measuring flow state in their entirety should be made cautiously, specifying which conditions are and are not met. We have made this recommendation explicit in the text (775-777) and in the checklist (514).

4. In the last part of the paper, the authors provide a summary of the points they raised and the specific points they challenged. This summary, however, also shows the limitations of the critique. Specifically, the authors state that their goals were to challenge the notion that flow...:

4. 1) ... is a unidimensional state.

From my knowledge of the literature, I never encountered an article/chapter in which a scholar considered flow as a unidimensional state. Each researcher acknowledges that flow is a complex state with multiple facets.

4 Thank you for your comments regarding our critique of the assumption that flow is unidimensional. We have removed our original comment about unidimensionality that you have raised and have instead, included a new, more explanatory section titled “Using self-report flow scales” (727-777). In this section, we explain that:

(a) The first way researchers claim flow is unidimensional is by explicitly stating it. We have cited this literature (741-742).

(b) The second way researchers claim flow is unidimensional is by averaging out multidimensional data to create global composite flow scores. Fig. 4 (714) illustrates the problems with reducing flow to a single score, especially when it comes to seeking neural correlates of such a complex state with different experiential components. We have also dedicated two paragraphs to evidence showing that flow is multidimensional and that the different dimensions contribute to flow with unequal weights (735-760).

(c) To ensure readers have ways to measure flow multidimensionally, we recognise and cite research that already adopts our views by analysing flow. These studies analysed the individual dimensions of flow against neurophysiological data instead of global composite flow scores (764) and are included in our checklist (520).

5. 2)... Can be operationalized via challenge-skill balance only.

Although it is true that flow many studies have used this approach, it is not clear whether researchers truly generally assume that this is the only method. Moreover, in a paper that wants to address this topic, one would expect some very specific recommendations, but these do not seem to be present in this paper. Merely saying that there are other ways to measure something would not suffice in order to provide a useful contribution to the literature.

5 Thank you for pointing this out. We agree about the magnitude of this limitation and have thoroughly addressed it in the revision, elevating it to a primary topic of discussion. As previously mentioned in response to your comment (3), we have introduced a new section titled “Personalising challenge levels” and the corresponding figure (611-675). In addition, we have added the following changes:

In the first paragraph of “the concept of flow”, we introduce perspectives of researchers that claim challenge skill balance is sufficient alone, and perspectives of others that claim challenge skill balance should be paired with a key motivational antecedent. We have amended this sentence to address both approaches with corresponding citations (140-141).

To address your valid point that we have not provided specific recommendations, we have

refined some sections to emphasize the variety of challenges that can be experimentally manipulated within the challenge-skill balance antecedent. This discussion is woven into our broader framework on active autonomy and detailed in the section “Types of challenges underlying flow states” (392-499), previously titled “Theoretical perspectives on autonomy and flow”). In the last paragraphs of this section, we have provided suggestions about how these types of challenges might be experimentally manipulated (465-499).

At the arc between the Introduction and the start of the main paper body, we have commented that the article will elaborate on these issues about the different types of challenges that can be experimentally manipulated (247-249).

We have also integrated the isolated types of challenge types that we recommend exploring experimentally into the practical checklist (515).

6. 3)... Cannot be experienced by activity-naïve individuals.

It makes perfect sense that it is probably easier to find flow in people engaging in their favored and optimal activities. For example, chess players, dancers, scientists doing science, etc. I can't imagine there would be many researchers who disagree with that statement. But in this paper, there are several sides to that.

First, the authors do not provide any convincing discussion of previous empirical studies showing that flow cannot be induced in participants in a lab setting. They just used the arguments that the original conceptualization of flow was based on observing experts. What evidence do the authors have (other than anecdotal) that flow cannot be induced in the lab?

- 6 Thank you for your comments about the seemingly obvious assumption that flow will be easier for individuals who are already familiar with the task. We agree that this seems obvious and cited empirical evidence supporting this argument in the section: “Participant skills and motivation” (550-553).

We have elaborated on our criticism of neurophysiological studies recruiting participants without any pre-established relationship with the tasks at hand (531-534).

In addressing the need for well-designed inclusion criteria and study designs that enhance intrinsic motivation, we have detailed best practices in the subsequent portion of this section and the checklist items (517). We also recommend using the SIMS (Guay et al., 2000) to screen for intrinsic motivation and mitigate the potential confounding effects of participants who may participate out of obligation rather than interest, such as students for course credits (578-581).

Regarding your concerns about our stance on the induction of flow in the lab, we have reviewed the manuscript and confirm that we have not directly made this claim. However, we do highlight that laboratory conditions might present obstacles to flow. Our approach, rather than attempting to rectify the inherent limitations of the lab environment, focuses on ensuring that other modifiable factors (e.g., motivation, personalising challenge, testing isolated challenge types, increasing intrinsic motivation, processing self-report items) are carefully integrated into research designs to facilitate flow.

7. Do the authors think that this would only apply to flow or do they consider validity a general issue with experimental research?

7. This is an intriguing question, leading us to articulate within the manuscript why flow research is particularly vulnerable to certain challenges, unlike much other experimental research. Our rationale for emphasizing the uniqueness of flow research stems from the distinct motivational and skill-based antecedents for experiencing flow (583-585). We argue throughout the manuscript that flow arises in familiar situations that people care about and are skilled in, and thus, experimental research on flow must ensure these conditions are met. This requirement contrasts with fields like motor learning or perception research, where the significance of task motivation or expertise may not be as critical. Nonetheless, the generalisability of lab-based findings to 'real-world' motor learning or perception is also being questioned. For example, in motor learning recent

experiments are attempting to use computer game players to collect data from experts over longer time periods that may better reflect the learning mechanisms that are more prominent in 'real world' motor learning (Listman et al. 2021).

Listman, J. B., Tsay, J. S., Kim, H. E., Mackey, W. E., & Heeger, D. J. (2021). Long-Term Motor Learning in the "Wild" With High Volume Video Game Data. *Frontiers in Human Neuroscience*, 15, 777779. <https://doi.org/10.3389/fnhum.2021.777779>

8. 4)... can occur over shorter periods

The authors mention the findings of two studies. In one study with a game, flow brain activity seemed to occur after 25 minutes (but how about the experience of flow?). In the other study, they refer to a study with piano players in which flow indicators seemed to occur after 90 minutes. In what way is that convincing? Would the authors maintain the view that, in principle, it is impossible (or very unlikely) that a skilled piano player can play a short song (say somewhere between 5 and 1 minutes) while in a flow.

8. Thank you for your comments on the impact of duration on experimental flow research. The scarcity of information on the effects of duration on flow states indeed presents a significant gap in current research. Given the nature and purpose of this perspective format, our goal is to raise awareness about the opportunity to test the effects of duration, since it has not been an explicit target of experimentation. In response to your comments, we have added duration as a manipulatable experimental variable that deserves empirical support in our checklist (515). We have also highlighted that due to the lack of targeted empirical investigation, this is currently an exploratory endeavour ("In the procedural duration section" (607-609)).

28th May 24

Dear Oliver

I have editorially evaluated the revisions of your Perspective titled "In search of authentic flow: Challenges in neurophysiological studies" and I am delighted to say that we are happy, in principle, to publish it in *Communications Psychology* under a Creative Commons 'CC BY' open access license.

We will not send your revised paper for further review if, in the editors' judgement, the referees' comments on the present version have been addressed. If the revised paper is in *Communications Psychology* format, in accessible style and of appropriate length, we shall accept it for publication immediately. I have attached an edited version of your manuscript, and ask you to attend to each comment in detail.

EDITORIAL REQUESTS:

* Please review the changes in the attached copy of your manuscript, which has been edited for style, and address the comments and queries I have added. If using Word, please use the 'track changes' feature to make the process of accepting your manuscript more efficient.

* Please check whether your manuscript contains third-party images, such as figures from the literature, stock photos, clip art or commercial satellite and map data. If any of the display items in your manuscript (figures, tables, boxes or movies) include images that are the same as, or are adaptations of, previously published images, please fill in the Third Party Rights Table, and return to us when you submit your revised manuscript. This information will enable us to obtain the necessary rights to re-use such material. If we are unable to obtain the necessary rights to use or adapt any of the material that you wish to use, we will contact you to discuss alternative options.

* *Communications Psychology* uses a transparent peer review system. On author request, confidential information and data can be removed from the published reviewer reports and rebuttal letters prior to publication. If you are concerned about the release of confidential data, please let us know specifically what information you would like to have removed. Please note that we cannot incorporate redactions for any other reasons.

*If you have not done so already, please alert me to any related manuscripts from your group that are under consideration or in press at other journals, or are being written up for submission to other journals (see www.nature.com/authors/editorial_policies/duplicate.html for details).

FORMATTING GUIDELINES:

You will find a complete list of formatting requirements following this link:

<https://www.nature.com/documents/commsj-style-formatting-checklist-review-perspective.pdf>

Please use the checklist to prepare your manuscript for final submission. In the following, I also highlight some issues of particular importance.

SUBMISSION INFORMATION:

In order to accept your paper, we require the following:

* A cover letter describing your response to our editorial requests.

* A separate document detailing your point-by-point response to any issues raised by our referees (please include the referees' comments in this document).

* The final version of your text as a Word or TeX/LaTeX file, with any tables prepared using the Table menu in Word or the table environment in TeX/LaTeX and using the 'track changes' feature in Word.

* Production-quality versions of all figures, supplied as separate files. Photographic images should be 300 dpi in RGB format (.jpg, TIFF or native Photoshop format) and any labels/scale bars included in a separate layer from the image. Line art, graphs and schemes should be vector format (.ai, .eps, .pdf); Adobe Illustrator files are preferred and will minimize production time. Any chemical structures or schemes contained within figures should additionally be supplied as separate Chemdraw (.cdx) files.

At acceptance, the corresponding author will be required to complete an Open Access Licence to Publish on behalf of all authors, declare that all required third party permissions have been obtained.

Please note that your paper cannot be sent for typesetting to our production team until we have received this information; therefore, please ensure that you have this ready when submitting the final version of your manuscript.

ORCID

Communications Psychology is committed to improving transparency in authorship. As part of our efforts in this direction, we are now requesting that all authors identified as 'corresponding author' create and link their Open Researcher and Contributor Identifier (ORCID) with their account on the Manuscript Tracking System (MTS) prior to acceptance. ORCID helps the scientific community achieve unambiguous attribution of all scholarly contributions. For more information please visit <http://www.springernature.com/orcid>

For all corresponding authors listed on the manuscript, please follow the instructions in the link below to link your ORCID to your account on our MTS before submitting the final version of the manuscript. If you do not yet have an ORCID you will be able to create one in minutes.

IMPORTANT: All authors identified as 'corresponding author' on the manuscript must follow these instructions. Non-corresponding authors do not have to link their ORCIDs but are encouraged to do so. Please note that it will not be possible to add/modify ORCIDs at proof. Thus, if they wish to have their ORCID added to the paper they must also follow the above procedure prior to acceptance.

To support ORCID's aims, we only allow a single ORCID identifier to be attached to one account. If you have any issues attaching an ORCID identifier to your MTS account, please contact the Platform Support Helpdesk.

[link redacted]

We hope to hear from you within two weeks; please let us know if the process may take longer.

Best wishes,

Marika